# Early Surgical Care of Anticoagulated Hip Fracture Patients Is Feasible—A Retrospective Chart Review of Hip Fracture Patients Treated with Hip Arthroplasty within 24 Hours

**DOI:** 10.3390/jcm11216570

**Published:** 2022-11-05

**Authors:** Carlos Pankratz, Raffael Cintean, Dominik Boitin, Matti Hofmann, Christoph Dehner, Florian Gebhard, Konrad Schuetze

**Affiliations:** Department of Trauma-, Hand-, and Reconstructive Surgery, Ulm University, Albert-Einstein-Allee 23, 89081 Ulm, Germany

**Keywords:** hip fracture, anticoagulation, early surgical care, geriatric trauma

## Abstract

Anticoagulative medication such as antiplatelet drugs (PAI, acetylsalicylic acid and direct platelet aggregation inhibitors), vitamin-K-antagonist Warfarin (VKA) or direct oral anticoagulants (DOAC) are common among hip fracture patients, and the perioperative management of these patients is a rising challenge in orthopaedic trauma. Our objective was to determine the effect of oral anticoagulation in patients receiving early endoprosthetic treatment within 24 h after their admission. For the period from 2016 to 2020, a retrospective chart review of 221 patients (mean age 83 ± 7 years; 161 women and 60 men) who were treated either with hemi- (*n* = 209) or total hip arthroplasty (*n* = 12) within 24 h after their admission was performed. We identified 68 patients who took PAI, 34 who took DOAC and 9 who took VKA medications. The primary outcome measures were the transfusion rate and the pre- and postoperative haemoglobin (Hb) difference. The secondary outcome measures were the in-patient mortality and the rate of postoperative haematomas that needed operative treatment. A logistic/ordinal regression was performed considering the related variables to prevent cofounding occurring. The mean time to surgery was significantly longer for the DOAC and VKA groups when they were compared to the controls (none 14.7 ± 7.0 h; PAI 12.9 ± 6.7 h; DOAC 18.6 ± 6.3 h; VKA 19.4 ± 5.5 h; *p* < 0.05). There was no difference in the preoperative Hb level between the groups. Overall, 62 patients (28%) needed blood transfusions during the in-patient stay with an ASA classification (*p* = 0.022), but the type of anticoagulative medication was not a significant predictor in the logistic regression. Anticoagulation with DOAC and grouped surgery times were positive predictors for a higher Hb difference in the patients who did not undergo an intraoperative blood transfusion (*n* = 159). Postoperative haematomas only occurred in patients taking anticoagulative medication (four cases in PAI group, and three cases in DOAC group), but the logistic regression showed that the anticoagulative medication had no effect. The in-patient mortality was significantly influenced by a high ASA grade (*p* = 0.008), but not by the type of anticoagulative medication in patients who were treated within 24 h. We conclude that the early endoprosthetic treatment of the anticoagulated hip fracture patient is safe, and a delayed surgical treatment is no longer justifiable.

## 1. Introduction

The demographic changes that are occurring in developing countries is inevitably associated with a rising number of hip fractures [1]. Until the present day, hip fractures have been associated with a high morbidity and mortality, as the 1-year mortality rates reach up to 20% [2]. Hip fracture patients are often multimorbid and suffer from various medical conditions such as cardiovascular diseases [3], with the indication for an anticoagulative medication. In 2020, already over 40% of all hip fracture patients in Germany had an anticoagulant in their long-term medication [4,5]. Consequently, the surgical treatment of the anticoagulated patient is a rising challenge for attending physicians. Most common among these are platelet aggregation inhibitors (PAI) such as Clopidogrel, Ticagrelor or acetylsalicylic acid (ASS) after a myocardial infarction or a stent implantation. The effect of these drugs lasts up to 7 days [6]. In parallel, the intake of direct oral anticoagulants such as Rivaroxaban, Dabigatran and Apixaban (DOAC) are becoming more and more popular. DOAC show shorter lasting effects and half-life times of 8–15 h. However, due to the renal and hepatic metabolism there are a wide range of individual elimination rates, especially in the frail geriatric patient [7,8]. The usage of special antidotes such as idarucizumab for dabigatran or andexanet for factor Xa inhibitors rivaroxaban and apixaban are still not a part of common clinical practice [9]. In contrast, the established vitamin-K-antagonist Warfarin (VKA) can be antagonised by vitamin K administration or prothrombin complex concentrates.

By pausing the drug intake and degradation, over time, each anticoagulant loses its effect, but a delayed surgical treatment of hip fractures is widely accepted as a major factor for having an extended hospital stay [10,11], increased mortality [12,13,14,15,16] and an elevated risk for postoperative complications such as pulmonary embolism, infections, renal failure, decubitus ulcers, acute pulmonary edema or myocardial ischemia [17,18,19]. Hip fractures are surgical emergencies, and there are clear international recommendations for an early surgery [20,21]. For example, the current S2 level guideline of the German traumatology society recommends the treatment of hip and proximal femur fractures within 24 h [22], whereas the optimal window of opportunity for surgery remains a subject of scientific discourse, and the surgery of anticoagulated patients is often reported to be delayed [23,24], and attending physicians find themselves caught between avoiding the unnecessary delay of surgical treatment and preventing serious bleeding complications.

Here, we hypothesise that the urgent arthroplasty of the anticoagulated hip fracture patient is safe and does not go along with an increased rate of red blood cell transfusions (RBCT) or significant postoperative haemoglobin differences. For this purpose, we retrospectively evaluated hip fracture patients who have been treated within 24 h by hip arthroplasty to investigate the effects of different oral anticoagulants on bleeding complications.

## 2. Materials and Methods

All of the presented data were obtained retrospectively with permission from the local ethical committee, and they were stored anonymously. Between January 2016 and December 2020, 431 patients were treated for an acute hip fracture in our level 1 trauma centre. We subsequently excluded patients who did not undergo a following surgical treatment due to death or them having a conservative treatment, patients with haematological disorders and the regular periodic need of RBCT, patients with blood clotting disorders, patients with additional injuries in need of surgical care and patients who were not operated within 24 h after their admission. The main reasons for surgery being performed after 24 h were an acute medical condition that needed to be treated before surgery, and organisational reasons, such as a lack of operating room capacity. After applying the exclusion criteria, 329 patients were identified. From the patients who were treated within 24 h, 221 received hip arthroplasty surgery and 108 patients received joint-conserving surgery (DHS or FNS, Co. DePuy Synthes, West Chester, PA, USA). Ultimately, we included 221 patients in the study, and from these, 209 received a dual head prothesis (CORAIL^®^ Hip System, SELF-CENTERING^TM^ Bipolar head, Co. DePuy Synthes, West Chester, PA, USA) and 12 received a total endoprosthesis (CORAIL^®^ Hip System, PINNACLE^®^ Acetabular Cup System, Co. DePuy Synthes, West Chester, PA, USA; Figure 1).

The patients were operated on while they were in a supine position using the modified lateral hip approach. All of the surgery procedures were performed or supervised by experienced orthopaedic trauma surgeons. The PAI were administered perioperatively, and they were not paused. The DOAC were stopped and bridged from the first postoperative day depending on patient’s specific thromboembolic risk with an intermediate-dose (enoxaparin 40 mg, 2 times a day) or high-dose (enoxaparin 1 mg/kg, 2 times a day) low-molecular-weight heparin. The VKA patients were treated with a vitamin K admission and had an INR control immediately before the surgery. If the INR was still over 1.5, prothrombin complex concentrates were administered. The VKA patients were postoperatively bridged depending on their INR and specific thromboembolic risk with an intermediate-dose or high-dose low-molecular-weight heparin. The DOAC and VKA treatments were resumed on postoperative day 8 if no bleeding complications had occurred.

We reviewed their clinical records—including the patient charts, blood values, anaesthesia protocols, surgery protocols and doctor’s letters. The primary outcome parameters were the intraoperative blood transfusion rate, the pre- to postoperative Hb difference and the postoperative haematoma requiring surgical revision during the in-patient stay. The indication of blood transfusion was made individually for every patient based on the factors such as Hb value < 8g/dL with accompanying clinicals symptoms such as hypotension, tachycardia, or vertigo. Here, only the red blood cell transfusions were considered. The preoperative Hb was measured at the point of admission; the postoperative Hb was measured on the first postoperative day. The secondary outcome parameter was in-patient mortality.

The data analysis was performed using SPSS (v25.0, Co. IBM, Armonk, NY, USA). The demographic characteristics are described as mean, standard deviation and range for the continuous data, and absolute and relative frequencies for the categorical data. For the categorial outcome measures, an logistic/ordinal regression was performed considering the related variables to prevent cofounding. Additionally, for the continuous outcome measures, a regression was performed including all of the possibly confounding variables. A *p*-value < 0.05 was considered as statistically significant.

## 3. Results

### 3.1. Patient Population

We reviewed medical records of 221 patients. General parameters such as age, gender, the American Society of Anesthesiologists Classification (ASA) and the type of anticoagulation were recorded (Table 1). For the statistical analysis, age (<80 years; ≥80 years), surgery time (<60 min; 60–90 min; >90 min) and time to surgery (type of anticoagulation) were divided into subgroups. The study cohort had a mean age of 83.2 ± 7.46 years, with 72.9% of them being female patients. The perioperative parameters such as AO/OTA fracture and dislocation classification, grouped operating time, grouped time to surgery and blood transfusion are listed in Table 2.

### 3.2. Time to Surgery

The overall mean time to surgery was 14.9 ± 7.0 h (Table 2). The anticoagulative treatment with DOAC (18.6 ± 6.3 h, *n* = 34) and VKA (19.4 ± 5.5 h, *n* = 9) increased the time to surgery when it was compared to the controls (14.7 ± 7.0 h, *n* = 110). In the case of the non-anticoagulated group which was compared to the PAI patient group (12.9 ± 6.7 h, *n* = 68), the differences were not statistically significant (*p* = 0.42; Figure 2).

### 3.3. Red Blood Cell Transfusion

Overall, 62 out of 221 patients, corresponding to 28%, received at least one RBCT (Table 2, Figure 3). The mean RBCT count was 0.55 ± 1.11 in the patients who did not take anticoagulative medication compared to 0.63 ± 1.03 in the PAI group, 0.82 ± 1.29 in the DOAC group and 0.33 ± 0.71 in the VKA group. A further analysis was performed by a logistic regression, which showed no dependency for the transfusion rate in relation to the type of anticoagulation, grouped age, AO fracture type or time to surgery. Only the ASA classification and the preoperative Hb level had a significant effect. In terms of the ASA, four patients had an increased risk for a red blood transfusion by a factor of 14.4 (*p* = 0.022). A lower preoperative Hb level significantly increased the risk for a red blood cell transfusion (*p* < 0.001).

### 3.4. Hb Level and Difference

There was no significant difference in the preoperative Hb levels in the patients who did and did not take anticoagulant medication (Figure 4). The postoperative Hb differences among the patients without the need for RBCT (*n* = 159) were investigated in dependence of their grouped age, AO fracture and dislocation classification, the type of anticoagulation, ASA classification, grouped surgery time and grouped time to surgery. The mean postoperative Hb difference in dependence of the anticoagulation treatment is shown in Table 3 and Figure 5. The logistic regression showed a significantly increased likelihood of higher Hb differences for the patients with higher preoperative Hb levels (*p* < 0.001), longer operating times (*p* = 0.001) and who took a DOAC treatment (*p* = 0.020). Without becoming statistically significant, the patients on VKA tended to a lower Hb difference when they were compared to the non-coagulated patients.

### 3.5. Postoperative Haematoma

Out of two hundred and twenty-one included patients, seven of them needed surgical revision due to a postoperative haematoma. Four cases occurred in the group of patients who took PAI, and three cases occurred in the patients who took DOAC. Because of the low occurrence of postoperative haematomas, a statistical analysis was performed by merging the different types of anticoagulation into one group. A logistic regression using the factors of grouped age (<80 years; ≥80 years), AO classification, ASA classification, grouped surgery time (<60 min; 60–90 min; >90 min), time to surgery and type of anticoagulation showed that only the operating time had a statistical tendency to determine the occurrence of postoperative haematoma (*p* = 0.086).

### 3.6. In-Patient Mortality

Out of 221 patients, 17 of them, relating to 7.7%, died during their in-patient stay. Of those seventeen deaths, nine (53%) occurred in the non-anticoagulation group, five (29%) occurred in the PAI group, two (12%) occurred in the DOAC group, and one (6%) occurred in the VKA group. The logistic regression showed that a high ASA classification was significantly associated with a higher mortality risk (*p* = 0.008), and as well as this, rising age tended to increase the mortality (*p* = 0.10).

## 4. Discussion

A prolonged time to surgery is an independent risk factor of mortality after a hip fracture [25], and it is associated with higher patient morbidity and mortality [19,26]. An ageing population correlates with a rising number of patients requiring anticoagulant medication. For these patients, their time to surgery must be balanced with the risk of the bleeding complications. In presented study, more than 50% of the patients treated with hip arthroplasty within 24 h had an anticoagulative treatment in their long-term medication. Attending physicians face the challenge to ensure optimal conditions for surgery and the patients’ safety regarding each type of anticoagulation therapy.

A treatment with PAI is the most common anticoagulative therapy. The mean wait until surgery in our PAI-study cohort was only 12.9 ± 6.7 h, and this did not differ among the non-coagulated patients. We could not observe an increased RBCT rate or Hb difference in the patients who were on PAI. These findings are in line with two meta-analyses examining the effect of clopidogrel [27,28]. Likewise, Collinge et al. reported no increased risk regarding the bleeding complication, transfusion rate and mortality after the surgical hip fracture treatment for patients on clopidogrel and ASS [29]. In all of the cited studies, the accepted delay until surgery was longer when it was compared to our mean time to surgery. Our results show that a surgical intervention should not be delayed in patients who are on PAI in terms of long-term medication.

The treatment with DOAC or VKA often expands the time to surgery for patients with an acute hip fracture [30,31,32,33]. Tran et al. reported a 40.0 h median time to surgery in patients who took DOAC or VKA compared to 26.2 h in the non-coagulated control group [24]. In our study, the mean time to surgery of patients on DOAC was 18.6 ± 6.3, and for patients on VKA, this was 19.4 ± 5.5 h. In contrast, the non-coagulated patients waited 14.7 ± 7.0 h until they underwent surgery. In our level I trauma department, acute hip fractures are treated as soon as possible regardless of the anticoagulation therapy. This study proves that there is still a delay to surgery for patients who are on DOAC and VKA. All of the patients were preoperatively prepared for intraoperative RBCT, and the patients on VKA were treated with vitamin K and prothrombin complex to achieve an INR that was less than 1.5. This preoperative coagulation optimisation might have further caused the observed prolonged time to surgery in patients who were on VKA.

The number of studies examining the complications and outcomes of anticoagulated patients receiving hip arthroplasty subsequent to experiencing an acute hip fracture is limited [34]. Overall, 28% of the patients in our study cohort needed at least one RBCT during or after surgery. However, we found no significant influence for the different types of anticoagulation on the RBCT rate. In accordance, Franklin et al., Schermann et al. as well as King et al. reported no difference in the blood transfusion rate and blood loss in the patients who were on DOAC who underwent early the surgical treatment of their hip fractures [30,31,35]. However, in all of the cited studies, the mean time to surgery for the anticoagulated patients outruns the critical 24 h mark after the patient’s admission. Franklin et al. communicated a 28.9 ± 11.8 h mean time to surgery for the patients who were on DOAC, but they observed the study cohort needed a hemiarthroplasty as well as a cephalomedullary nailing treatment. Schermann et al. even reported a 42.3 ± 27.3 h average time to surgery for the patients who were on DOAC. For the patients without an RBCT, the intake of DOAC was a significant risk factor for higher postoperative Hb differences. This finding may be explained by the reported shorter time to surgery when they were compared to the mentioned studies.

In case of VKA, Cohn et al. showed no difference in the transfusion rate and blood loss in the VKA-treated patients when they were compared to the non-coagulated control group [32]. In our study, the patients on VKA showed even lower Hb differences when they were compared to the non-coagulated patients. Still, the number of VKA patients was low, and the VKA treatment showed no significant effect in the logistic regression, but there are similar observations in the literature in the course of proximal femur fractures [36]. In our department, the blood coagulation of all of the patients who are on VKA is laboratory controlled and antagonised with vitamin K or prothrombin complex concentrate if these are needed. This preoperative optimisation of the clotting might cause these findings. It remains unclear if the substitution of vitamin K or prothrombin complex concentrate might also reduce the blood loss in patients who do not undergo an anticoagulation therapy.

A high ASA grade was associated with a higher risk for RBCT and in-patient mortality. There is already some evidence that for patients with severe comorbidities and minor dislocated fractures, arthroplasty might not be the best line of treatment [37,38]. A prolonged operating time had a significant effect on the Hb difference and tended to increase the occurrence of postoperative haematoma. This underlines the importance and need of specialised orthopaedic trauma surgeons in the care of hip fracture patients.

Postoperative haematoma with the need of surgical intervention was observed in seven cases. All of the cases were associated with an anticoagulative therapy with either PAI (*n* = 4) or DOAC (*n* = 3) in terms of long-term medication. Still, anticoagulative medication showed no significant effect in the logistic regression which might be explained by the low frequency of haematomas. In contrast, in the case of ASS, Deveraux et al. demonstrated in a large, randomised trial that the perioperative intake of ASS had no significant effect on mortality and the rate of myocardial infarction, but it increased the risk of surgical bleeding complications [39]. Other recorded surgical complications such as thrombosis, pulmonary embolisms, cardiac infarctions, stroke, pneumonia, urinary tract infections, acute renal failure and deep tissue infections were not increased. Here, we observed an in-patient mortality of 7.7% that was in line with the values that have been reported elsewhere [2,40,41,42]. Thereby, mortality was significantly affected by the ASA classification but not by the anticoagulative therapy.

Elsewhere, in the course of proximal femur fractures, a significant lower Hb level at the point of admission and an increased blood transfusion rate for the patients on DOAC is reported [36]. Here, in accordance with Schermann et al., for hip fractures (AO type 31B), no differences in the Hb level at the point of admission was observed [31]. This may arise from different anatomical conditions, as proximal femur fractures are often associated with injuries of femoral circumflex arteries, while in hip fractures, intracapsular fracture bleeding is often self-limiting. These divergent findings may indicate that fracture morphology is an independent risk factor and might one day result in different guidelines.

Due to the retrospective design of it, our study has inevitably some limitations. The time to surgery included the time span from admission to surgical incision. Thereby, the exact time from trauma and admission could not be recorded. Additionally, because of different admission times, the blood draw interval varies as the Hb difference was calculated from the preoperative and postoperative Hb levels on the first postoperative day. Unfortunately, the one-year mortality rate could not be evaluated and was only recorded a small number of patients who had a later readmission to our hospital. A follow-up examination should be, therefore, part of further prospective studies. Finally, from a methodical point of view, only the patients who were operated within 24 h after their admission were included in the study which may have led to a certain selection bias.

Our study advocates for a short time to surgery as the mean time to surgery was only 14.9 ± 7.0 h. In almost all of the comparable studies, the mean time to surgery was clearly over 24 h. Therefore, to our knowledge this is the first study which focusing on anticoagulated patients with hip fractures that were treated by hemiarthroplasty within the first 24 h after their admission. Confounding variables such as fracture classification, surgery time, patient age and ASA classification have been considered. The evaluated parameters such as RBCT rate, pre- to postoperative Hb difference, postoperative haematoma and in-patient mortality are verifiable and of highly clinical relevance.

Our findings support the current guidelines recommending the urgent surgical treatment of hip fracture patients. As a consequence of this and a former work [36], we created an interdisciplinary standard operating procedure for our department in order to keep he time to surgery for all patients as low as possible despite their anticoagulation therapy type since the delayed surgical treatment of the anticoagulated hip fracture patient is no longer justifiable.

## 5. Conclusions

Using a standard operating procedure, the early surgical care of hip fractures in patients who take an anticoagulative therapy is proven to be safe, and this showed no increased bleeding risk in this retrospective study. All of the patients, regardless of their type of anticoagulative therapy, should be prepared preoperatively for possible intraoperative transfusions.

## Figures and Tables

**Figure 1 jcm-11-06570-f001:**
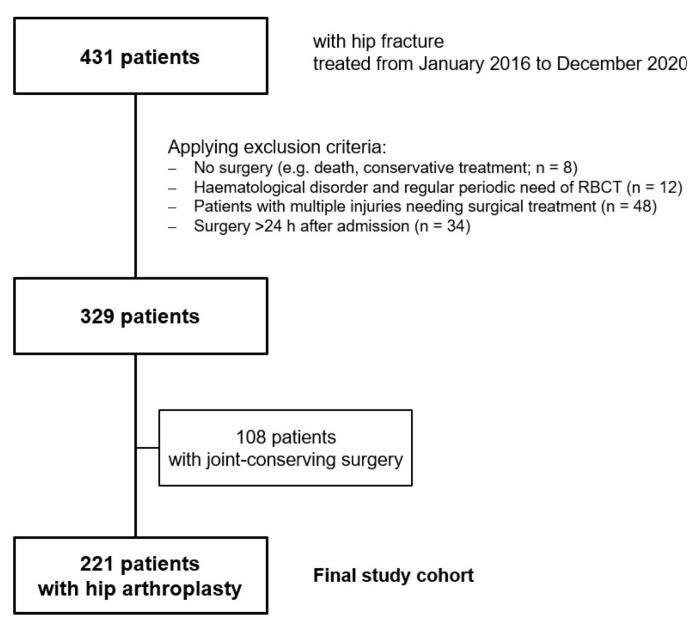
Study flow chart: a retrospective chart review.

**Figure 2 jcm-11-06570-f002:**
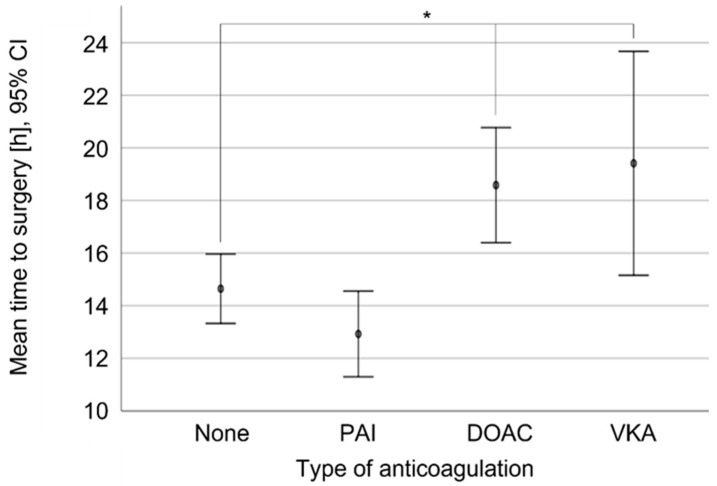
Mean time to surgery [h] depending on anticoagulative medication. PAI: Platelet aggregation inhibitors including acetylsalicylic acid; DOAC: Direct oral anticoagulants; VKA: Vitamin-K-antagonist Warfarin; *p* < 0.05 *.

**Figure 3 jcm-11-06570-f003:**
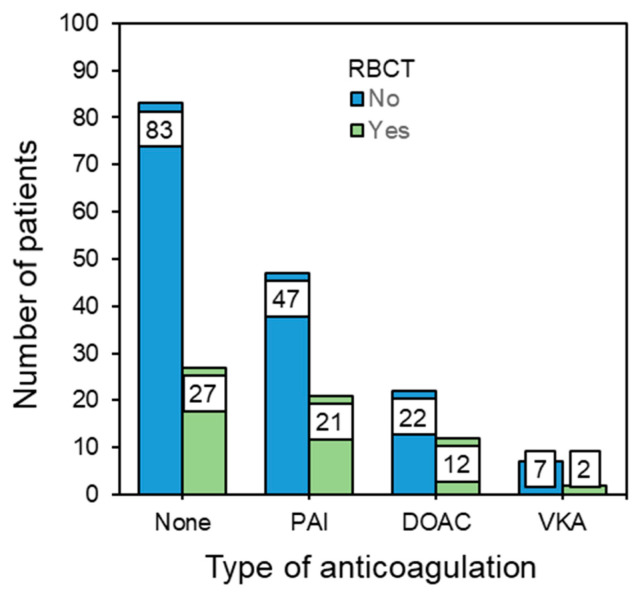
Red blood cell transfusion (RCBT) depending on anticoagulative medication. PAI: Platelet aggregation inhibitors including acetylsalicylic acid; DOAC: Direct oral anticoagulants; VKA: Vitamin-K-antagonist Warfarin.

**Figure 4 jcm-11-06570-f004:**
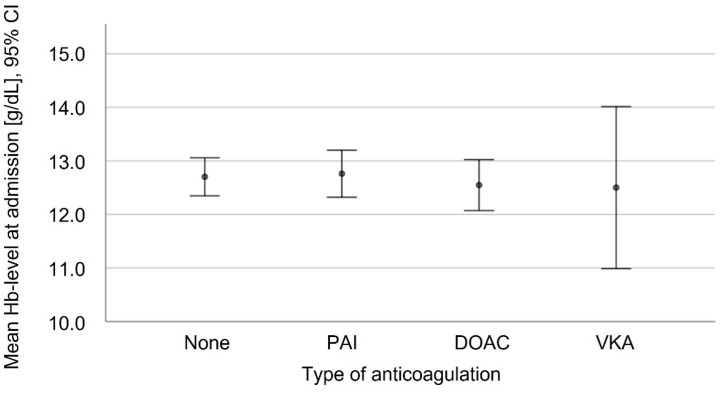
Hb levels at admission depending on anticoagulative medication. PAI: Platelet aggregation inhibitors including acetylsalicylic acid; DOAC: Direct oral anticoagulants; VKA: Vitamin-K-antagonist Warfarin.

**Figure 5 jcm-11-06570-f005:**
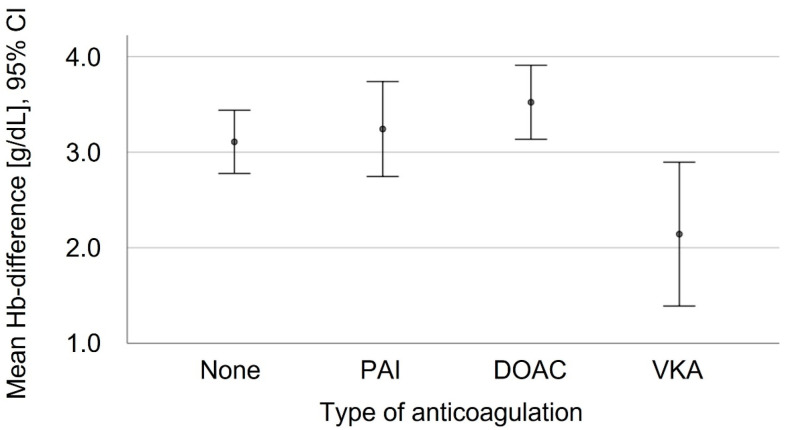
Mean postoperative Hb difference [g/dL] depending on different type of anticoagulation. Logistic regression showed only an increased risk for a higher Hb difference for patients taking DOAC. PAI: Platelet aggregation inhibitors including acetylsalicylic acid; DOAC: Direct oral anticoagulants; VKA: Vitamin-K-antagonist Warfarin.

**Table 1 jcm-11-06570-t001:** Patient population.

Variable	Mean/Count	SD/Percent
Age [years]	83.2	±7.5
<80	71	32.1%
≥80	150	67.9%
Gender		
male	60	27.1%
female	161	72.9%
ASA		
1	-	0%
2	22	10%
3	163	73.8%
4	36	16.3%
Type of anticoagulation		
none	110	49.8%
PAI	68	30.8%
VKA	9	4.1%
DOAC	34	15.4%

ASA: American Society of Anesthesiologists classification; PAI: Platelet aggregation inhibitors including acetylsalicylic acid; VKA: Vitamin-K-antagonist Warfarin; DOAC: Direct oral anticoagulants.

**Table 2 jcm-11-06570-t002:** Perioperative factors.

Variable	Count/Mean	%/Range
AO fracture classification		
31B1	37	16.7%
31B2	149	67.4%
31B3	35	15.8%
Operating time [min]	79	27–235
Grouped operating time		
<60 min	43	19.5%
60–90 min	119	53.8%
>90 min	59	26.7%
Time to surgery [min]	895	92–1438
none	879	92–1438
PAI	776	150–1437
VKA	1165	400–1436
DOAC	1115	191–1436
RBCT		
Yes	62	28.1%
No	159	71.9%

PAI: Platelet aggregation inhibitors including acetylsalicylic acid; VKA: Vitamin-K-antagonist Warfarin; DOAC: Direct oral anticoagulants; RBCT: red blood cell transfusion.

**Table 3 jcm-11-06570-t003:** Postoperative Hb differences among patients who did not take RBCT (*n* = 159).

Variable	Mean	SD
Postoperative Hb differences [g/dL]	3.16	1.49
none	3.11	1.51
PAI	3.24	1.69
DOAC	3.52	0.87
VKA	2.14	0.81

PAI: Platelet aggregation inhibitors including acetylsalicylic acid; DOAC: Direct oral anticoagulants; VKA: Vitamin-K-antagonist Warfarin.

## Data Availability

The data presented in this study are available on reasonable request from the corresponding authors. The data are not publicly available due data protection reasons.

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
