# Peer review of "Early Surgical Care of Anticoagulated Hip Fracture Patients Is Feasible—A Retrospective Chart Review of Hip Fracture Patients Treated with Hip Arthroplasty within 24 Hours"

_jcm, 2022, doi:10.3390/jcm11216570_

Round 1

Reviewer 1 Report

The study design is presented in the Introduction, but the purpose of the study is not explicit.

In Materials and Methods, the type of study must be clearly and completely presented. It should the sample, but it is not clear the type of sample. Is it a probabilistic or non-probabilistic sample? And among them, which type of sampling?

Also in materials and methods, it should make explicit reference to the variables under study.

The results are presented in detail and the discussion is adequate. The limitations of the study could be improved with methodological aspects, namely those related to sampling.

The conclusion could be more detailed in addressing the study's objective. It does not present the implications arising from this study (for practice, research and education.

It used 21 references from the last 5 years (from 2017) and 18 with 5 or more years. Could have more recent references (less than 5 years)

Author Response

First, we want to thank you for your constructive comments and possibility to improve our work.

Point 1: The study design is presented in the Introduction, but the purpose of the study is not explicit.

Response 1: We have revised the introduction and specified the study purpose.

Point 2: In Materials and Methods, the type of study must be clearly and completely presented. It should the sample, but it is not clear the type of sample. Is it a probabilistic or non-probabilistic sample? And among them, which type of sampling?

Response 2: We revised and specified. This is a retrospective study. For the period January 2016 to December 2020, we included all hip fracture patients treated with arthroplasty and operated within 24 h after admission. Hence, it is a non-probability sampling.

Point 3: Also in materials and methods, it should make explicit reference to the variables under study.

Response 3: We revised materials and methods section.

Point 4: The results are presented in detail and the discussion is adequate. The limitations of the study could be improved with methodological aspects, namely those related to sampling.

Response 4: We revised discussion section and added an annotation to possible selection bias.

Point 5: The conclusion could be more detailed in addressing the study's objective. It does not present the implications arising from this study (for practice, research and education.

Response 5: We revised conclusion section.

Point 6: It used 21 references from the last 5 years (from 2017) and 18 with 5 or more years. Could have more recent references (less than 5 years)

Response 6: We revised cited literature and added more recent reference if possible. As we tried to use primary sources some literature is older than 5 years but not outdated since available literature is still limited.

Kind regards.

Reviewer 2 Report

Dear Author, 

congratulations on a great paper.

The paper is very well written with clear text. The main question that has been addressed is operating femoral neck fractures as soon as possible even in the patients that are on some kind of tromboprophylaxis. The topic is as relevant as it can be because of the substantial socioeconomic burden that femoral neck fracture in eldery population poses for the healthcare system. Clear arguments (relatively low complication rate of haematomas and revisions) corroborate the conclusion that the operation should not be prolonged. 

I have just one issue to be cleared. What were the criteria for total hip arthroplasty vs dual head prosthesis? Was there any difference between these subgroups?

Best regards 

Author Response

Point: What were the criteria for total hip arthroplasty vs dual head prosthesis? Was there any difference between these subgroups?

Response: Thank you for your positive feedback. The subgroup of patients who underwent total hip arthroplasty were predominantly younger, healthy patients (< 75 years) without the option of joint-conserving surgery. Statistical matching of these two subgroups showed no statistically significant differences. However, the small group size (n = 12) should be considered.

Kind regards.

Reviewer 3 Report

Dear Authors,

congratulations on your research idea. Your work deals with a subject of much debate. I recommend minor revisions.

Thank you.

1. Introduction: There are some controversies about the influence of the time to surgery on mortality. Some international study results are contradictory.

Add a new paragraph to the introduction and point out the clear international recommendations for an early surgery. This will emphasize the importance of your work. 

You may find some reference in the document (https://iqtig.org/downloads/auswertung/2020/17n1hftfrak/QSKH_17n1-HUEFTFRAK_2020_QIDB-RR-P_V01_2019-12-16.pdf) page 5-7.

2. L 47 "Duration of action"? Better: "Drug effect"

3. L 49-51 avoid convoluted style: Please change into two sentences 

4. L 73 Inappropriate wording: "Exclusion criteria included". Please change.

5. L 73-74 "patients, who ultimately did not have surgery" Explain this fact in more detail. Do you mean patients with indication for conservative therapy? palliative situation? deceased in emergency room?

6. L74 "patients with the need of regular RBCT" Explain this fact in more detail. Do you mean patients with haematological disorders and regular periodic need of RBCT?

7. L74-75 "patients in whom hip fracture was not the only diagnosis in need of surgical care" Explain this fact in more detail. Do you mean patients with additional diagnosis which requires a treatment of urgent priority?

8. Did you exclude patients with genetic blood-clotting disorders (like hemophiliacs,...)

9. L 75 "patients not operated within 24 h after admission" Why did you exclude this cohort? Isn't it a useful control group? Please give a statement.

10. L 73-78 Complement the particular numbers of excluded patients. Maybe a table or chart would be useful.

11. L 98 and 103 "postoperative haematoma " It is named in the section primary AND secondary outcome. Please modify. 

12. L 102 "measured ad admission" ="measured at admission"

13. Describe the surgical procedure in detail. Is it standardized? Approach for prothesis? Do you use tranexamic acid or other hemostatic agents? Do you apply compression bandage?

14. Your hospital seems to have a strict and standardized concept of early surgery < 24h. You need to point this out more expressly. Your approach goes far beyond the scope of the current medical guidelines.

15. Please create an short and catchy graph or diagram with the interdisciplinary standard operating procedure of your department for patients with anticoagulation. This could serve as a blueprint for other hospitals and could influence the authors of future medical guidelines.

16. L 116 "Error! Reference source not found." Repetitive finding. Resolve the error.

17. L 116-117 "For statistical analysis age (< 80 116 years; ≥ 80 years)" Why did you chose this grouping? preliminary work? comparable literature? 

18.  L 271-291 Change the order of paragraphs: Limitations first, study benefits last.

19. L 293: Please add: Using a standard operating procedure, early surgical care of hip fractures in patients with anticoagulative therapy is proved  to be safe...

20. L 296 Point out that this work is very useful as complement to actual medical guidelines. 

Author Response

First, we want to thank you for your constructive comments and possibility to improve our work.

Point 1: Introduction: There are some controversies about the influence of the time to surgery on mortality. Some international study results are contradictory.

Response 1: Added.

Point 2: L 47 "Duration of action"? Better: "Drug effect"

Response 2: Changed.

Point 3: L 49-51 avoid convoluted style: Please change into two sentences

Response 3: Changed.

Point 4: L 73 Inappropriate wording: "Exclusion criteria included". Please change.

Response 4: Changed.

Point 5: L 73-74 "patients, who ultimately did not have surgery" Explain this fact in more detail. Do you mean patients with indication for conservative therapy? palliative situation? deceased in emergency room?

Response 5: Changed.

Point 6: L74 "patients with the need of regular RBCT" Explain this fact in more detail. Do you mean patients with haematological disorders and regular periodic need of RBCT?

Response 6: Changed.

Point 7: L74-75 "patients in whom hip fracture was not the only diagnosis in need of surgical care" Explain this fact in more detail. Do you mean patients with additional diagnosis which requires a treatment of urgent priority?

Response 7: Changed.

Point 8: Did you exclude patients with genetic blood-clotting disorders (like hemophiliacs,...)

Response 8: Yes, but the patient cohort did not include any patient with a known genetic blood-clotting disorder.

Point 9: L 75 "patients not operated within 24 h after admission" Why did you exclude this cohort? Isn't it a useful control group? Please give a statement.

Response 9: We focussed only on patients undergoing surgery within 24 h after admission. Patients without anticoagulant in long-term medication served as the control group. Adding patients outside of the 24 h interval would miss our intension, as the anticoagulant, especially for DOAC, should be in most cases already partly metabolised. The comparability regarding bleeding complications would suffer and the influence of other confounding factors such as existing comorbidities may come to the fore.

Point 10: L 73-78 Complement the particular numbers of excluded patients. Maybe a table or chart would be useful.

Response 10: We added a chart.

Point 11: L 98 and 103 "postoperative haematoma " It is named in the section primary AND secondary outcome. Please modify.

Response 11: Changed.

Point 12: L 102 "measured ad admission" ="measured at admission"

Response 12: Changed

Point 13: Describe the surgical procedure in detail. Is it standardized? Approach for prothesis? Do you use tranexamic acid or other hemostatic agents? Do you apply compression bandage?

Response 13: Added. All patients were operated in supine position by modified lateral hip approach. Tranexamic acid or other hemostatic agents are not part of standard procedure. We don’t apply any compression bandage.

Point 14: Your hospital seems to have a strict and standardized concept of early surgery < 24h. You need to point this out more expressly. Your approach goes far beyond the scope of the current medical guidelines.

Response 14: See Response 15.

Point 15: Please create a short and catchy graph or diagram with the interdisciplinary standard operating procedure of your department for patients with anticoagulation. This could serve as a blueprint for other hospitals and could influence the authors of future medical guidelines.

Response 15: For several years, supported by available literature, our department is treating femoral neck and trochanteric fractures very progressive (See also Schuetze, K.; Eickhoff, A.; Dehner, C.; Gebhard, F.; Richter, P.H. Impact of oral anticoagulation on proximal femur fractures treated within 24 h – A retrospective chart review. Injury 2019, 50, 2040–2044, doi:10.1016/j.injury.2019.09.011). The results of presented and above cited work has led to development of an in-house SOP for the anticoagulated geriatric trauma emergency patient. But this workflow was not existing when the presented data was collected. Hence, we would prefer to present our interdisciplinary operating procedure after implementation and validation.

Point 16: L 116 "Error! Reference source not found." Repetitive finding. Resolve the error.

Response 16: We revised and renewed cross-references. The error message could not be reproduced.

Point 17: L 116-117 "For statistical analysis age (< 80 116 years; ≥ 80 years)" Why did you chose this grouping? preliminary work? comparable literature?

Response 17: We chose this grouping as it corresponds approximately to mean age (83.7 years) and the resulting group sizes allowing a well-founded statistical evaluation.

Point 18: L 271-291 Change the order of paragraphs: Limitations first, study benefits last.

Response 18: Changed.

Point 19: L 293: Please add: Using a standard operating procedure, early surgical care of hip fractures in patients with anticoagulative therapy is proved to be safe...

Response 19: Added.

Point 20: L 296 Point out that this work is very useful as complement to actual medical guidelines.

Response 20: Added.